**www.cambridge.org/qrd**

# On the micelle formation of DNAJB6b

Andreas Carlsson[1] , Ulf Olsson[2] and Sara Linse[1]

[1]Biochemistry and Structural Biology, Chemical Centre, Lund University, Lund, Sweden and [2]Physical Chemistry, Chemical Centre, Lund University, Lund, Sweden

self-association; aggregation; oligomers; chaperone action; affinity

**Corresponding author:**
Andreas Carlsson;
Email: andreas.carlsson@biochemistry.lu.se

## Abstract

The human chaperone DNAJB6b increases the solubility of proteins involved in protein aggregation diseases and suppresses the nucleation of amyloid structures. Due to such favourable properties, DNAJB6b has gained increasing attention over the past decade. The understanding of how DNAJB6b operates on a molecular level may aid the design of inhibitors against amyloid formation. In this work, fundamental aspects of DNAJB6b self-assembly have been examined, providing a basis for future experimental designs and conclusions. The results imply the formation of large chaperone clusters in a concentration-dependent manner. Microfluidic diffusional sizing (MDS) was used to evaluate how DNAJB6b average hydrodynamic radius varies with concentration. We found that, in 20 mM sodium phosphate buffer, 0.2 mM EDTA, at pH 8.0 and room temperature, DNAJB6b displays a micellar behaviour, with a critical micelle concentration (CMC) of around 120 nM. The average hydrodynamic radius appears to be concentration independent between ∼10 μM and 100 μM, with a mean radius of about 12 nm. The CMC found by MDS is supported by native agarose gel electrophoresis and the size distribution appears bimodal in the DNAJB6b concentration range ∼100 nM to 4 μM.

CAMBRIDGE
UNIVERSITY PRESS

## Introduction

The human chaperone DNAJB6b has been found to suppress amyloid formation of several amyloidogenic peptides, including polyglutamine peptides (Hageman *et al.,* 2010; Gillis *et al.,* 2013; Månsson *et al.,* 2014*b*; Kakkar *et al.,* 2016; Rodríguez-González *et al.,* 2020; Thiruvallu-van *et al.,* 2020), amyloid β peptide (Månsson *et al.,* 2014*a*, 2018; Österlund *et al.,* 2020), α-synuclein (Aprile *et al.,* 2017; Deshayes *et al.,* 2019; Arkan *et al.,* 2021), TDP43 (Udan-Johns *et al.,* 2014) and IAPP (Chien *et al.,* 2010). DNAJB6b is expressed in many tissues and organs, including the brain, and is present in both the cytosol and the nucleus, whereas the isoform DNAJB6a is only found in the nucleus. The two isoforms share 231 residues, out of 241 for DNAJB6b, and 326 for DNAJB6a (Hanai and Mashima, 2003). DNAJB6b has two folded domains connected by a flexible linker region, which contains about half of the residues in the protein. The C-terminal domain mainly consists of a four-stranded β-sheet, whereas the N-terminal domain, called a J-domain, adopts a mostly α-helical structure and is a common feature of DNAJ proteins (Hageman *et al.,* 2010; Kampinga *et al.,* 2019). A structure model of the intact DNAJB6b, predicted using Alpha Fold (Jumper *et al.,* 2021; Mirdita *et al.,* 2022), is shown in Fig. 1.

A striking feature of many chaperones is their self-assembly into relatively large oligomers (Schonfeld *et al.,* 1995; Hageman *et al.,* 2010; Baldwin *et al.,* 2011; Månsson *et al.,* 2014*b*). An understanding of chaperone action and the formation of chaperone-client co-assemblies thus requires a detailed description of self-association equilibria. Oligomers of wide size distribution have been observed for DNAJB6b (Månsson *et al.,* 2014*b*; Karamanos *et al.,* 2019), and considerable effort has been made to characterise the interactions in the self-assemblies (Söderberg *et al.,* 2018; Karamanos *et al.,* 2019; Karamanos *et al.,* 2020). A more detailed description of the self-assembly process is crucial for understanding the physicochemical properties of DNAJB6b in solution, and in the extension its propensity to interact with amyloid-prone clients. A high chemical potential of DNAJB6b may drive its interactions with amyloid-forming peptides, as well as its self-assembly behaviour (Linse *et al.,* 2021). Formation of oligomers has been reported for several other chaperones, such as αB crystalline (Baldwin *et al.,* 2011), DNAJB8 (Hageman *et al.,* 2010; Ryder *et al.,* 2023) and Hsp70 (Schonfeld *et al.,* 1995).

The current work concerns the self-assembly of DNAJB6b in solution. The main questions addressed in this work are as follows: *(i)* Does the DNAJB6b self-assembly correspond to an equilibrium micelle formation, characterised by an equilibrium size distribution, or should it rather be seen as a precipitation of a dense protein phase? *(ii)* Is there a characteristic (critical) DNAJB6b concentration marking an onset of aggregation? *(iii)* to what extent is the self-assembly process perturbed by a covalently attached fluorophore at the C-terminus? As shown in Fig. 1, the fluorophores used in this study, Alexa647 and IRdye680, add about 1 kDa to the 26.9 kDa DNAJB6b and have a mixed hydrophobic and charged character.

Microfluidic Diffusional Sizing (MDS) is most commonly used to study interactions between different proteins. The technique relies on differences in diffusion rates for various particle sizes. See Methods for a more detailed description. In this work, it is used to examine equilibrium sizes, question *(i)*, and onset of aggregation, question *(ii)*. To answer question *(iii)*, the protein was studied both with and without a C-terminally attached Alexa647. Agarose gel electrophoresis under native conditions was conducted to complement the MDS technique, providing a visual evaluation of the size distribution as a function of protein concentration, using IRdye680-labelled protein.

## Methods

### DNAJB6b expression and purification

DNAJB6b was expressed, purified, and fluorescently labelled according to previously published protocol (Linse, 2022). Alexa647 (Thermo Fisher Scientific) coupled to an added cysteine residue at DNAJB6b C-terminus was used in the MDS, in combination with non-labelled wild-type DNAJB6b. In the agarose gel electrophoresis, the fluorophore used was IRdye680 (LI-COR Biosciences), attached in the same manner as Alexa647.

### DNAJB6b sample preparation

The buffer for all experiments was 20 mM sodium phosphate, 0.2 mM EDTA, at pH 8.0, filtered through a wwPTFE-filter (0.22 μm pore size). All chemicals were of analytical grade. All samples were prepared, stored, and measured at room temperature. It was noted that upon thawing the frozen purified protein, the solution had a milky appearance, with strong light scattering, indicating the presence of large precipitates. The sample clears over a few hours at room temperature, and the visible light scattering disappears. Thus, all protein solutions were thawed at least 1 day prior to usage.

### MDS

The MDS technique (see Fig. 2 for a schematic description) is based on particle diffusion across parallel laminar flows in a microfluidic system to obtain the diffusion coefficient, which is used to calculate a hydrodynamic radius (Yates *et al.*, 2015). The flows are separated into two chambers at the end of the channel, and the particle concentrations in each chamber are measured using fluorescence detection. The sample can either be pre-labelled with a covalently attached fluorophore, or post-labelled with a suitable probe, for

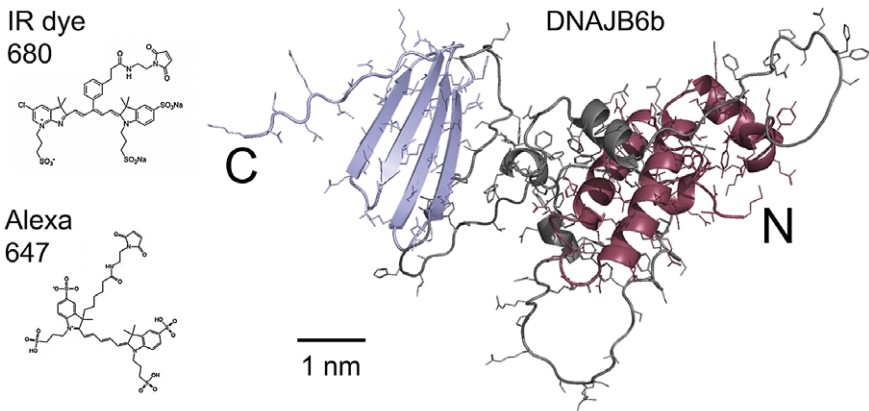

**Figure 1.** Right: model of the DNAJB6b structure as predicted using Alpha Fold, with the N-terminal domain coloured red, the C-terminal domain blue and the linker region grey. Left: chemical structures of the two fluorophores IRdye680 (Li-Cor, 2023), and Alexa647 (Gebhardt *et al.*, 2021) shown at the same scale as the DNAJB6b model.

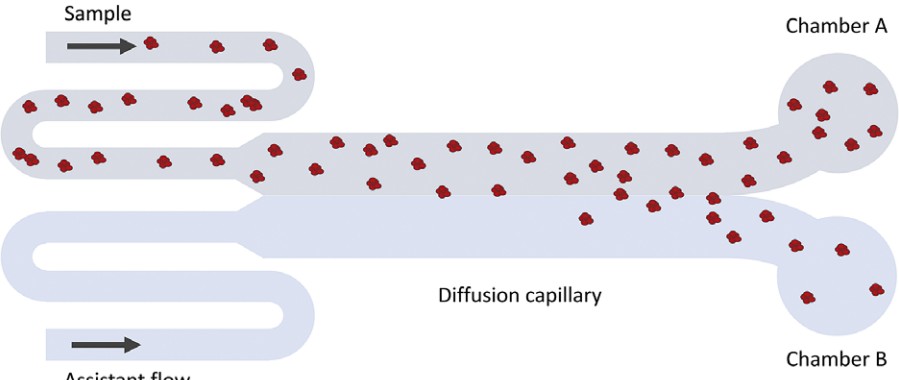

**Figure 2.** Schematic illustration of the MDS technique. Fluorescence detection is used to measure the fraction of protein that has ended up in chamber A versus B. Since only diffusion causes particles to move in the cross-flow direction, the intensity ratio between chamber A and B can be related to the average hydrodynamic radius of the particles.

instance, ortho-phthalaldehyde (OPA). In the former case, the limit of detection is lower than 20 nM. In the latter case, the concentration range for reliable detection was found to be around 100 nM–28 μM for DNAJB6b.

The translational diffusion coefficient, $D$, can be obtained from the concentration ratio between chamber A and B. Using the Stokes-Einstein relation, Eq. (1), it is possible to translate $D$ to a hydrodynamic radius, $R_H$. Here, $k_B$ is the Boltzmann factor, $T$ is the absolute temperature and $\eta$ is the solvent viscosity:

$$D = \frac{k_B T}{6\pi\eta R_H} \tag{1}$$

The protein needs to diffuse in the capillary for a sufficient period of time for the method to distinguish between various sizes. Given the large difference in size between monomeric DNAJB6b and its highest oligomeric forms, different flowrates were used for samples in different concentration ranges, to cover a large interval of detectable sizes. The instrument Fluidity One M (Fluidic Analytics, Cambridge UK) permits 5 flowrates (suggested radius in parenthesis): setting 1 (1–4.7 nm), setting 2 (2–9.3 nm), setting 3 (3–14 nm), setting 4 (3.7–17 nm), and setting 5 (4.3–20 nm).

### Surface-area-to-volume ratio

DNAJB6b was found to be surface active, meaning that some protein will be depleted from the solution to cover the surfaces of the sample container. A decrease in concentration due to surface adsorption was found to be critical in the nM-range, where a significant fraction of the total protein is adsorbed at the surfaces. The effect was examined for 20 nM Alexa647-DNAJB6b by measuring the total fluorescence intensity using MDS (Fluidity One M from Fluidic Analytics, Cambridge UK) as a function of time in contact with the container. Two sample volumes were analysed, 100 μL and 5 mL (both in Eppendorf protein low-binding 5 mL tubes), providing highly different surface-area-to-volume ratios.

### Equilibration time

Alexa647-DNAJB6b was diluted from 1.4 μM to 20 nM at time zero, to a sample volume of 5 mL in a 5-mL Eppendorf protein low-binding tube. MDS (instrument Fluidity One M from Fluidic Analytics, Cambridge, UK) was used to obtain an average hydrodynamic radius at various time points after dilution, ranging from 15 minutes to 11 days. Flowrate settings 2 and 3 were used at each time point, measuring four replicates in each case. The protein solution was kept in dark between measurements.

To examine the dissociation kinetics of non-labelled DNAJB6b, the instrument Fluidity One (Fluidic Analytics, Cambridge UK) was used, which uses the post-labelling setup with *ortho*-phthalaldehyde (OPA) as a fluorescence probe after diffusion. Due to limitations in the detectable concentration range, these samples were diluted from 6 μM to 100 nM. Three samples were diluted and measured at various time points (30 minutes to 5 days), to obtain triplicates during the first day. In the following days, six measurements were used for each timepoint. The flowrate used was suitable for 2–20 nm particles.

### Concentration-dependent size

At concentrations below 2 μM, 1 mL samples were prepared in 5 mL Eppendorf protein low-binding tubes, with minimal tilting of the tubes to obtain minimal surface-to-volume ratio. At concentrations

higher than or equal to ∼2 μM, the adsorption to surfaces can be neglected compared to the total amount of protein in each sample, making it sufficient to work with smaller volumes (50 μL or more were used). For the detection using Alexa647-protein (using the instrument Fluidity One M, Cambridge UK), all samples were mixed with 20 nM labelled DNAJB6b and various amounts of non-labelled DNAJB6b to total concentrations between 20 nM and 100 μM. The samples were kept in dark between measurements. Samples at 6 μM or lower were measured 5 and 9 days after dilution. For the higher concentrations, no dilutions were done, so the equilibration time was less of an issue. These samples were left to equilibrate between 3 and 7 days prior to measurement. Flowrate settings 2 and 3 were used for samples of 20 nM–6 μM DNAJB6b. Flowrate settings 2, 3 and 4 were used for 1–6 μM, and flowrate settings 4 and 5 were used for 16–100 μM. For the post-labelling setup using instrument Fluidity One, the measurements were performed 3 days after dilutions. For this instrument, the flowrate suitable for 2–20 nm radius particles was used.

### Native agarose gel electrophoresis

Non-labelled DNAJB6b at various concentrations was mixed with IRdye680-labelled DNAJB6b at constant concentration, so that all samples contained 5 nM labelled protein and the total protein concentration varied from 5 nM to 32 μM, in steps of about a factor of 2. The agarose gel was prepared from 1% w/v SEAkem LeAgarose (Lonza Bioscience, Basel, Switzerland) in 50 mM Hepes/NaOH pH 8.5 and cast on the hydrophilic side of a 110 x 205 mm Gelbond film (Lonza Bioscience, Basel, CH). 10 μL sample was loaded per lane and the gel was operated horizontally on a water-cooled bed at 260 V for 25 min (Johansson, 1972) The gel was blotted onto a PVDF membrane using a 5 kg weight as a press. The membrane was dried and scanned using an IR fluorescence scanner Odyssey CLx, Li-Cor (Bad Homburg, Germany). The electrophoretic mobility was measured as the length travelled during the electrophoresis, in arbitrary length units (l. u.).

Using the image analysing software, ImageJ, the IR-intensities for each concentration were analysed along the direction of electrophoretic movement.

## Results

### Equilibration time

The measurements of concentration-dependent size and the onset of aggregation require samples at equilibrium. Thus, as a first stage of studying the self-assembly of DNAJB6b, the time to reach equilibrium upon dilution was examined, that is how long time is needed before the particle size does not change any longer. The average hydrodynamic radius, $\langle R_H \rangle$, was obtained using MDS at various times after dilution, both in the case of non-labelled DNAJB6b and Alexa647-DNAJB6b, Fig. 3a. Single exponential decay functions are used to fit the data for each series. Regarding the non-labelled DNAJB6b, the time until $\langle R_H \rangle$ appears to be time independent is ∼3 days, and for the Alexa647-labelled protein ∼5 days.

In Fig. 3b, the total fluorescence intensities are shown for two samples diluted from 1.4 μM to 20 nM, as a function of time after dilution, during which the samples were kept in the dark. The volumes of the two samples were 5 mL and 100 μL, respectively, resulting in highly different surface-area-to-volume ratios. The fluorescence intensity of the smaller volume sample drops during

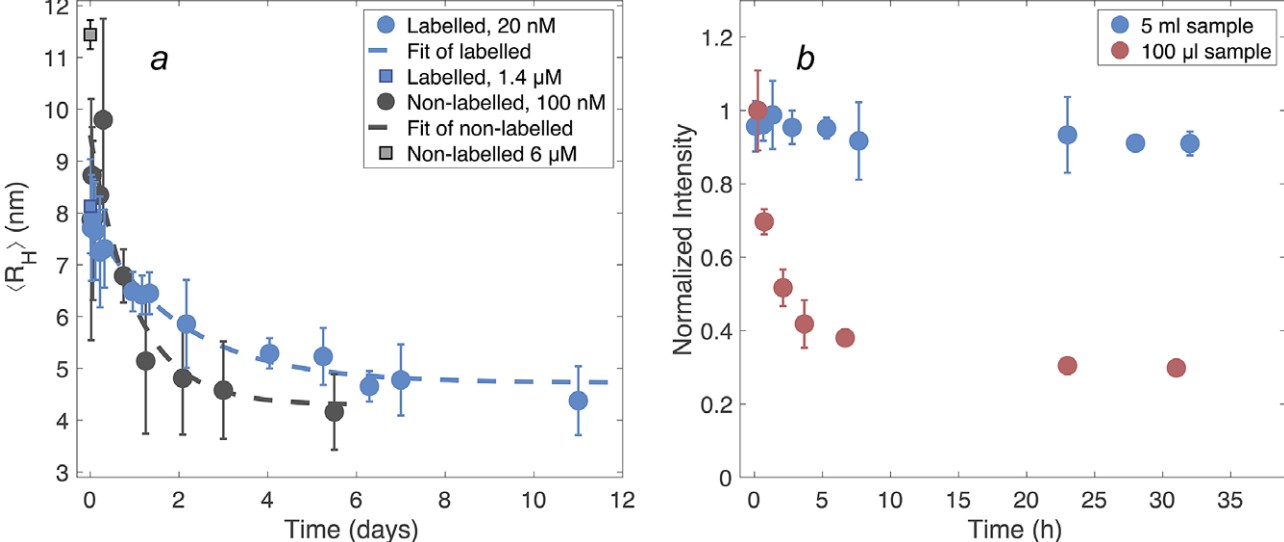

**Figure 3.** (*a*) $\langle R_H \rangle$ of DNAJB6b at various times after dilution, obtained using MDS. The blue series represents Alexa647-labelled protein diluted from 1.4 µM to 20 nM, measured in quadruplicates for each flowrate (setting 2 and 3) and time. The standard deviations are shown as errorbars. The non-diluted sample is shown as a blue square at time zero. Non-labelled DNAJB6b (black), diluted from 6 µM to 100 nM, was measured in a minimum of triplicates for each time point. The grey square represents the $\langle R_H \rangle$ of the non-diluted sample. Exponential decay functions (dashed lines) on the form $f(x) = a * e^{-t/b} + c$ are fitted to each series. The labelled protein is described by a = 3.2 nm, b = 2.3 days, and c = 4.7 nm. Non-labelled protein is described by a = 5.2 nm, b = 1.0 days, and c = 4.3 nm. (*b*) Total fluorescence intensities of 20 nM Alexa647-DNAJB6b during 31 h after dilution, measured with MDS. The sample volumes used during storage in each container were 5 mL (blue) and 100 µL (red), providing two highly different surface-area-to-volume ratios.

the first 5 hours to about 40% of the initial intensity, while the sample of the larger volume shows no significant intensity decay.

### Concentration-dependent size

After having established the time needed to reach consistent results, the self-assembly behaviour of DNAJB6b was studied over a broad range of concentrations, using MDS at pH 8.0, room temperature, and modest ionic strength. Fig. 4 shows the $\langle R_H \rangle$ versus protein concentration, with linear (panel *a*) and logarithmic (panel *b*) x-axes. The data shown in blue represent samples with 20 nM Alexa647-DNAJB6b and varying amount non-labelled protein to final concentrations ranging between 20 nM and 100 µM. Samples of solely non-labelled protein were measured in the range 100 nM–28 µM (black) to assess whether the labelled protein interacts in assemblies of similar sizes as the non-labelled protein. We find that the data for the two series follow each other closely.

The measured size appears to plateau above ~10 µM. The mean radius of all samples above 10 µM is 12 ± 0.5 nm. In the logarithmic plot, a concentration-independent region below 100 nM can be seen. The mean radius here is 4.9 ± 0.2 nm. Linear fitting of the radius versus the logarithm of the concentration in the range 160 nM–2 µM, together with a line at 4.9 nm, are plotted in red in panel *b*. The lines intersect at 120 nM.

### Electrophoretic mobility

The electrophoretic mobility of DNAJB6b was examined as a function of protein concentration, in native agarose gel electrophoresis. An IR fluorophore, IRdye680, covalently attached to DNAJB6b, was used to visualise the protein in the gel. The concentration of labelled protein was kept constant at 5 nM, and the amount non-labelled protein was varied to obtain total concentrations of 5 nM to 32 µM. The blotted gel is shown in Fig. 5, panel *a*. By analysing the IR fluorescence intensities in the direction of

movement, the electrophoretic mobility profiles can be compared in more detail. In panel *b*, the profiles are plotted with the shifted *y*-axis, superimposed with the gel. The electrophoretic mobility varies from 30 ± 5 length units (l. u.) for the lowest concentrations, to less than 20 l. u. for the highest concentrations. Profiles of samples in the range 37 nM–4 µM are plotted with the same *y*-axis in panel *c*, to examine how the population with lower electrophoretic mobility (around 20 l. u.) increases with protein concentration and how the population with higher electrophoretic mobility (around 30 l. u.) decreases with protein concentration. 131 nM is the lowest concentration at which a population with lower mobility is visible.

### Discussion

The present work addresses the self-assembly of DNAJB6b, mainly regarding *(i)* equilibrium states, *(ii)* onset of aggregation, and *(iii)* the effect of a covalently attached fluorophore at the C-terminus. Since fluorophore-labelled proteins were used to address both *(i)* and *(ii)*, we first evaluate aspect *(iii)*. Using MDS, $\langle R_H \rangle$ were obtained for various DNAJB6b concentrations, both with and without a covalently attached Alexa647 fluorophore at the C-terminus of the protein. The overlapping data in Fig. 4 imply that the labelled protein assembles with the non-labelled protein without major changes in aggregation numbers. Thus, samples with a low concentration of Alexa647-DNAJB6b represent the wild-type protein well in regards of the average self-assembly sizes. However, a difference in the dissociation kinetics was observed (Fig. 3), suggesting that the attachment changes the protein properties to some extent. This is not surprising, since Alexa647 contributes with both charge and hydrophobicity, as implied by its chemical structure, shown in Fig. 1. A closer description of the exchange rates between oligomers and their subunits will be a topic for future studies. We note that Söderberg *et al.* (2018) detected cross-links between DNAJB6b monomers of different isotopes, after mixing oligomers for 1 h (one time point studied), and the current time

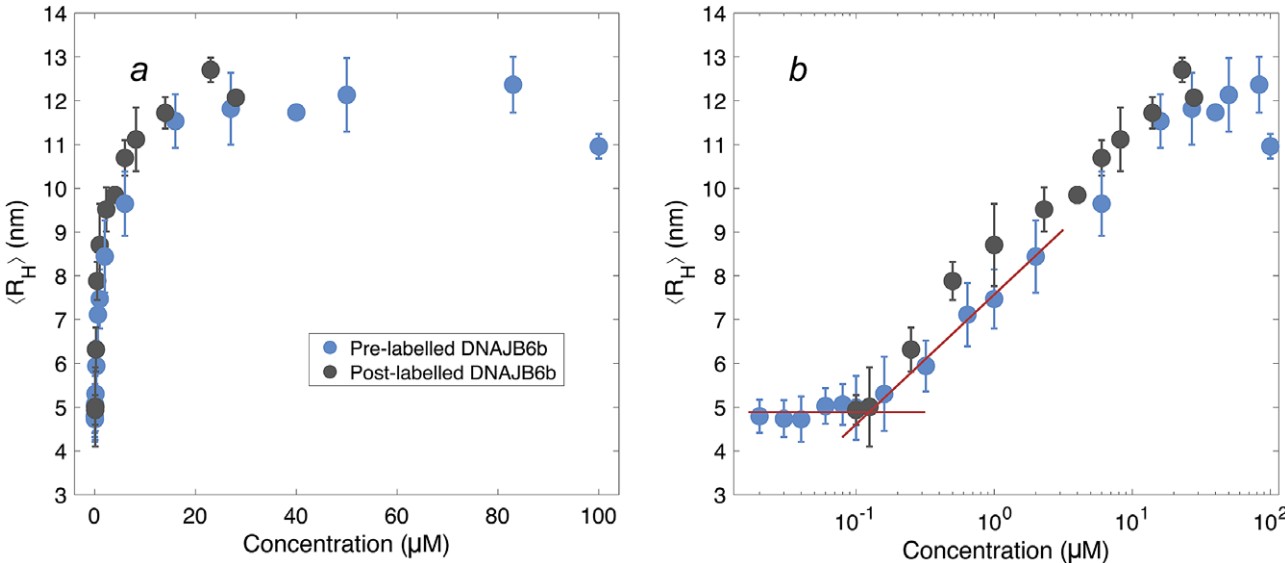

**Figure 4.** $\langle R_H \rangle$ of DNAJB6b measured with MDS as a function of protein concentration, plotted with linear (*a*) and logarithmic (*b*) *x*-axis. Blue data points show measurements for samples containing 20 nM Alexa647-DNAJB6b and varying amount of non-labelled protein to total concentrations ranging between 20 nM and 100 μM. Each data point represents a mean $\langle R_H \rangle$ of several flowrates and times since dilution. For more details regarding data collection, see Methods and Supplementary material. In black, non-labelled DNAJB6b at 100 nM–28 μM. All samples were measured in at least triplicates (except for 28 μM non-labelled protein, which is a singlicate). The errorbars represent the standard deviations of the replicates. The red lines describe the concentration independent region below 100 nM and a linear fit of the radius versus the logarithm of the concentration in the range 160 nM–2 μM. The two lines intersect at 120 nM.

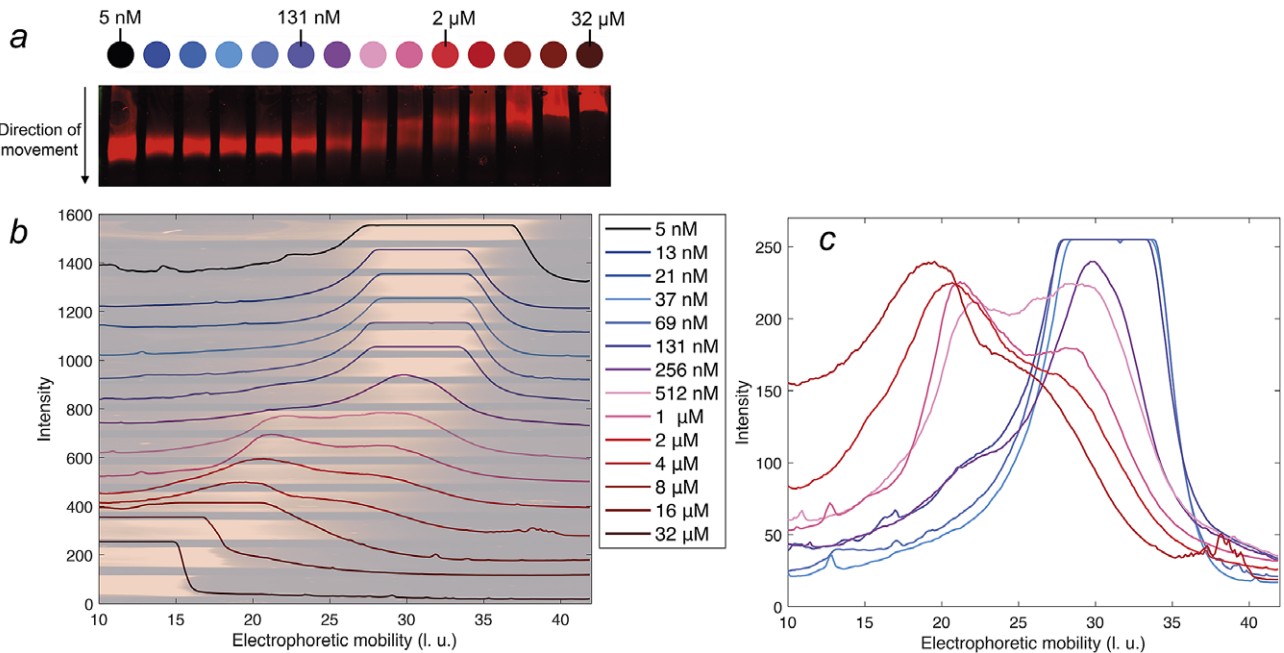

**Figure 5.** Native agarose gel electrophoresis of 5 nM IR-labelled DNAJB6b together with various concentrations of non-labelled DNAJB6b, with total concentrations as indicated by different colours, see full legend in panel *b*. Panel *a* shows a scan of the blotted gel, obtained using an IR-fluorescence scanner. Analysis of the IR intensities along the direction of movement gives an electrophoretic mobility profile for each sample, in arbitrary length units, l. u. All profiles are plotted in panel *b* with shifted baselines. The profiles are superimposed with the gel, in pale colors, rotated and expanded along the direction of the applied voltage relative to panel a, so that pixels and electrophoretic mobility are directly translatable. In panel *c*, the profiles of 37 nM–4 μM DNAJB6b are plotted with a common *y*-axis, to compare the profiles where the shift in electrophoretic mobility occurs.

course data suggest that longer time is needed to reach an equilibrium size distribution.

The overall MDS data strongly indicate that the average aggregate sizes, presented in Fig. 4, represent equilibrium values, question (*i*). These $\langle R_H \rangle$ data were obtained after long equilibration times; 5–9 days for concentrations below 6 μM, and 3–7 days for concentrations above 6 μM. This, together with the fact that the $\langle R_H \rangle$ values in Fig. 4 show a clear and reproducible trend, including a constant plateau ($\langle R_H \rangle$ = 12 ± 0.5 nm) at higher concentrations, implies that the observed aggregation corresponds to an

equilibrium micelle formation, where micelles at a given concentration are characterised by an equilibrium size distribution.

The lowest DNAJB6b concentration at which the micelles are present to a notable extent, question *(ii)*, called the critical micelle concentration (CMC), is estimated from Fig. 4. It may be noted that this is the lowest concentration where an increase in size is detected using MDS, but it does not necessarily mean that no oligomers are present at lower concentrations, but to a much smaller extent. In other words, we do not know how "critical" it is, and various techniques may provide different estimates of this concentration, depending on the observed property and the sensitivity of the technique. The CMC obtained using MDS (Alexa647-labelled series) is estimated to 120 nM, from the intercept of the lines describing the mean $\langle R_H \rangle$ at 20–100 nM and a linear fit of radius versus the logarithm of concentration in the range 160 nM–2 μM, as shown in Fig. 4. It is also possible to estimate a CMC from the native agarose electrophoresis (Fig. 5), given that electrophoretic mobility decreases with increasing aggregation number. At 131 nM, the first sign of change in electrophoretic mobility is detected, implying a CMC below 131 nM. Thus, the two techniques used in this work provide similar CMC estimates of DNAJB6b, despite the highly different measuring methods.

From the MDS technique, the $\langle R_H \rangle$ is obtained for each sample, but as implied from the agarose gel electrophoresis, and as previously reported (Månsson *et al.,* 2014*b*; Karamanos *et al.,* 2019), DNAJB6b assembles into a wide size distribution. From the current work, aggregation numbers or quantitative size distributions of the assemblies cannot be accurately estimated, since more knowledge about the assembly densities, structure, and shape, are needed. These issues will be addressed in future studies. However, it can be noted that the electrophoretic mobility profiles, seen in Fig. 5, are not well described by single Gaussian functions between 131 nM to 4 μM, but rather by bimodal distributions.

Is DNAJB6b mostly monomeric or dimeric when it is not part of a micelle? This is an important question since it might be vital to know which form of DNAJB6b is involved in the amyloid-suppressing mechanism. But it is not trivial to estimate the hydrodynamic radii of potential oligomeric states since the monomeric protein to a large extent is intrinsically disordered. A lower size limit of DNAJB6b is estimated to $R_H = 2.0$ nm for a monomer, and $R_H = 2.5$ nm for a dimer. These limits are calculated by considering a well-folded spherical 26.9 kDa protein and using a molar specific volume of 0.72 mL/g, a mean value which is calculated from data of various folded proteins listed in Cantor and Schimmel (1980) and van Holde *et al.* (1998). Upper size limits can be estimated by considering completely unfolded proteins, that is random coils, with the number of residues, $N = 241$ in the monomeric case and $N = 482$ in the dimeric case. The radius of gyration of such molecules can be described by $R_g = R_0 N^\nu$ nm, where experiments and simulations provide $R_0 = 0.2$ nm and $\nu = 0.60$ as a good generalisation for unfolded proteins (Wilkins *et al.,* 1999; Kohn *et al.,* 2004; Tran *et al.,* 2005). Furthermore, $R_H$ for a random coil can be obtained from $R_g$ via the relation $R_H \approx 2/3 R_g$ (Richards, 1980). This approach provides the upper limits $R_H = 3.6$ nm for monomers and $R_H = 5.4$ nm for dimers. Our experimentally obtained $\langle R_H \rangle$ from MDS is 4.9 ± 0.2 nm. This is larger than the calculated upper limit of a monomer, suggesting that DNAJB6b may be dimeric to a substantial extent at 20–100 nM, or at least not completely monomeric.

In conclusion, the DNAJB6b oligomerisation appears to be an equilibrium attribute, corresponding to a micellar behaviour, at pH 8, modest ionic strength and room temperature. The onset of aggregation (CMC) was found to be around 120 nM, when evaluated with MDS and native agarose gel electrophoresis. The $\langle R_H \rangle$ of DNAJB6b at equilibrium was found to vary with concentration, ranging from 4.9 ± 0.2 nm below CMC to 12 ± 0.5 nm above ∼10 μM. DNAJB6b with a C-terminally attached Alexa647-fluorophore assembles together with non-labelled DNAJB6b, seemingly without affecting the $\langle R_H \rangle$.

**Open peer review.** To view the open peer review materials for this article, please visit http://doi.org/10.1017/qrd.2023.4.

**Acknowledgements.** We would like to thank Cecilia Emanuelsson, Lund University, and Tuomas Knowles, University of Cambridge, for stimulating scientific discussions. Jelica Milosevic, Lund University, contributed to the purification of DNAJB6b, for which we are grateful.

**Supplementary material.** The supplementary material for this article can be found at http://doi.org/10.1017/qrd.2023.4.

**Author contribution.** A.C., U.O. and S.L. conceived and designed the study. A.C. and S.L. conducted data gathering. A.C., U.O. and S.L. performed data analyses. A.C. wrote the article with input from the other authors.

**Financial support.** This study was funded by grant 2015–00143 from the Swedish Research Council (S.L.), 2022.0059 from the Knut & Alice Wallenberg Foundation (S.L., U.O.), and 101097824 from the European Research Council (S.L.).

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
