## [Reviewer Report]

*Comments to Author*: This is a very interesting paper. I strongly believe that the investigation of the mechanisms of self-oligomerization of anti-aggregation chaperones has a lot to teach us about how they perform their functions. That is not an easy task in biophysical terms and therefore the manuscript is timely and should be of high interest to the field. The experiments are well designed and performed thoroughly, I especially enjoyed Figure 4 as I think it represents an advance on DNAJB6b oligomerization. I only have a few suggestions for improvement:

1. The results in Figure 3 were slightly surprising to me as the re-equilibration of DNAJB6 oligomers back to a stable oligomer population is taking place over a longer timescale than I would expect. That must be the outcome of a pretty slow subunit exchange rate within the oligomers. Soderberg et al (doi: 10.1038/s41598-018-23035-9) observed subunit exchange in mixed isotopic cross-linking experiments within 1 hour. Would that be compatible with the data of Figure 3? Is it possible to estimate the subunit exchange rate from the MDS data? The authors mention in their discussion that this may be the topic of a future paper but if they could add a comment about it in the current manuscript I would find it beneficial.

2. The subsection titles in the results could be improved to be more specific and different from those in the methods section.

3. Although I get what the authors are trying to say in lines 198 -203 in the Discussion, I think this paragraph could be made more clear.

4. The introduction could be expanded in order to further highlight the importance of studying chaperone oligomerization. Perhaps a few sentences about how MDS works would also beuseful.

---

## [Reviewer Report]

*Comments to Author*: Reviewer #1: This is a very interesting paper. I strongly believe that the investigation of the mechanisms of self-oligomerization of anti-aggregation chaperones has a lot to teach us about how they perform their functions. That is not an easy task in biophysical terms and therefore the manuscript is timely and should be of high interest to the field. The experiments are well designed and performed thoroughly, I especially enjoyed Figure 4 as I think it represents an advance on DNAJB6b oligomerization. I only have a few suggestions for improvement:

1. The results in Figure 3 were slightly surprising to me as the re-equilibration of DNAJB6 oligomers back to a stable oligomer population is taking place over a longer timescale than I would expect. That must be the outcome of a pretty slow subunit exchange rate within the oligomers. Soderberg et al (doi: 10.1038/s41598-018-23035-9) observed subunit exchange in mixed isotopic cross-linking experiments within 1 hour. Would that be compatible with the data of Figure 3? Is it possible to estimate the subunit exchange rate from the MDS data? The authors mention in their discussion that this may be the topic of a future paper but if they could add a comment about it in the current manuscript I would find it beneficial.

2. The subsection titles in the results could be improved to be more specific and different from those in the methods section.

3. Although I get what the authors are trying to say in lines 198 -203 in the Discussion, I think this paragraph could be made more clear.

4. The introduction could be expanded in order to further highlight the importance of studying chaperone oligomerization. Perhaps a few sentences about how MDS works would also beuseful.